# Implementation of R&D Results and Industry 4.0 Influenced by Selected Macroeconomic Indicators

**Jena Švarcová** [1,*] **, Tomáš Urbánek** [2] **, Lucie Povolná** [1] **and Eliška Sobotková** [1]

[1] Department of Economics, Tomas Bata University in Zlín, Zlín 76001, Czech Republic; povolna@utb.cz (L.P.); sobotkova@utb.cz (E.S.)

[2] Department of Statistics and Quantitative Methods, Tomas Bata University in Zlín, Zlín 76001, Czech Republic; turbanek@utb.cz

* Correspondence: svarcova@utb.cz



**Featured Application: An economic recession can destroy the potential of INDUSTRY 4.0 projects and R&D projects. Current EU grant support, tax relief, and other specific factors appear to be more important for the development of new R&D projects in the Czech Republic than the effects of economic recession.**

**Abstract:** Successful timing of INDUSTRY 4.0 projects in businesses can be disrupted by the coming of a recession. The authors assume a close link between INDUSTRY 4.0 and research and development (R&D) projects. R&D projects are statistically internationally monitored and have a significant impact on European Union economic policies. This article explores the impact of the two economic recessions in 2009 and 2012–2013 on the number of R&D entities and human resources involved in R&D in the Czech Republic. The method of multivariate statistics with dummy variables was used. Research has shown that different sectors (business sector, government sector, higher education sector, and non-profit sector) show a different development of the number of R&D entities in times of economic crisis. The research findings indicate that current European Union grant support, tax relief, and other specific factors appear to be more important for the development of R&D projects in the Czech Republic than the effects of economic recession. In terms of longer time horizons, however, the effects of the business cycle cannot be ignored. In order to predict economic development, enterprises and other subjects can use leading macroeconomic indicators.

**Keywords:** INDUSTRY 4.0; economic recession; research and development indicators

## 1. Introduction

A number of authors, such as Weyer et al. [1] or Wang et al. [2], recognize INDUSTRY 4.0 as one of the major factors in improving manufacturing processes, increasing productivity and economic performance. At the same time, an economic crisis may be one of the most important factors that could lead to the failure of promising INDUSTRY 4.0 projects, as Povolná and Švarcová [3] point out. Pavelková et al. [4] examines economic performance in the Czech Republic (CR) in the dominant automotive sector in the pre-crisis, crisis, and post-crisis periods. There is clear pressure on the introduction of INDUSTRY 4.0 and research and development (R&D) knowledge. Statistical data in the CR shows that the number of subjects actively involved in R&D did not decline during two recessions in 2009 and 2012–2013, but even increased in selected sectors (the business sector). This is a completely untypical phenomenon. Usually, in times of recession, gross domestic product (GDP) and the number of economically active entities decrease in the economy under review. Macroeconomic performance (measured by GDP) and number of economically active entities increase at a time of boom. This article

tested hypotheses of whether the recessions affect the number of subjects actively involved in R&D, both in the business sector and in the government sector, universities, and non-governmental non-profit organizations in the CR. At the same time, the development of workforces in R&D workplaces is examined. The results show that the impact of economic recession on the R&D entities in the CR is not significant nowadays and the economic crisis does not affect the number of subjects evolving INDUSTRY 4.0 and R&D. Subjects of R&D in Europe are much more influenced by other factors, for instance grants to support R&D from European Union funds. The Czech Republic is very closely linked to the German economy, which is the main driver of the European Union (EU) economy, so it can be expected that the results in the Czech Republic will be similar to those in Germany. At the same time, it should be emphasized that the support system for R&D projects is uniform across the EU, so similar impacts can be expected in other EU countries. It is a stimulus for economic policy and state support and transnational grant programs (such as EU programs), to increase the supply of subsidies, grants, and tax relief in times of economic crisis. Active support for R&D can, thus, be one of the counter-cyclical measures to reduce the negative effects of economic crises.

This paper is arranged as follows. We first introduce some literature review and related work in Section 2. Section 3 explains the materials and methods. Section 4 describes the results. Section 5 presents discussion and conclusions in detail, and Section 6 highlights actual INDUSTRY 4.0 successes in Central Europe, brings final conclusions of the paper, provides corresponding policy suggestions, and describes the limitations.

## 2. Literature Review

Conceptual framework questions: Is the INDUSTRY 4.0 phenomenon beneficial in both developed and emerging economies? Which areas does INDUSTRY 4.0 focus on? Does INDUSTRY 4.0 have negative impacts? What is the macroeconomic significance of INDUSTRY 4.0 for the Czech Republic, and in a wider international context? Compared to other countries in the world, do the Czech Republic and its business partner Germany need to revitalize the Information and Communication Technologies (ICT) industry and support INDUSTRY 4.0? Can new technologies be the only source of economic growth or is human resource development and its long-term impact important? In the short term, it is important for the economy to monitor the signals of coming economic recessions, which are provided by the statistical indicators IFO Business Climate Index (leading indicator for economic activity in Germany prepared by the Ifo Institute for Economic Research in Munich, Germany) and ZEW indicators (The ZEW – Leibniz Centre for European Economic Research in Mannheim is an economic research institute and monthly publish their forecasts for macroeconomic indicators). Can we use R&D data to describe the INDUSTRY 4.0 phenomenon?

Dalenogare et al. [5] highlights evidence of INDUSTRY 4.0's benefits, not only in advanced economies, but also in emerging economies. In their extensive study, they examined and discussed the contextual conditions of the Brazilian industry that may require a partial implementation of the INDUSTRY 4.0 concepts created in developed countries.

Diez-Olivan et al. [6] concludes that one of the main objectives of data science in the context of INDUSTRY 4.0 is to effectively predict abnormal behavior in industrial machinery, tools, and processes, so as to anticipate critical events and damage that eventually cause significant economic losses and safety issues. Many authors focus on the review of essential standards and patent landscapes for the Internet of Things (IoT) and industrial IoT, such as Xia et al. [7] and Trappey et al. [8]. Some authors emphasize the relevance of visual computing for manufacturing processes, for example, Lee et al. [9] focus on service innovation and smart analytics using big data as a key enabler for next generation advanced manufacturing. Rajput and Singh [10] show that the IoT ecosystem and IoT big data are the most influential IoT enablers that help industry practitioners effectively implement INDUSTRY 4.0.

Kovacs [11], in his article, The Dark Corners of INDUSTRY 4.0, emphasizes that INDUSTRY 4.0 processes have not only positive economic impacts but also negative side-effects on the environment and society. It is a serious topic that needs to be addressed so that negative impacts are minimized.

Implementation of INDUSTRY 4.0, taking into account influences on the environment and society, provides immense opportunities for the realization of sustainable manufacturing [12].

The macroeconomic significance of INDUSTRY 4.0 is being studied both in the context of the Czech Republic and in a wider international context. The initial conditions of aspects of process management in the context of company strategies in Czech enterprises are evaluated by Tuček [13]. Pavelková et al. [4] examines economic performance in the Czech Republic in the dominant automotive sector in the pre-crisis, crisis, and post-crisis periods. There is clear pressure on the introduction of INDUSTRY 4.0 and R&D knowledge. Increasing the productivity and economic performance of companies using INDUSTRY 4.0 is becoming an important part of the competitive struggle and more companies are trying to use R&D results in business processes. Kang et al. [14] brings comparisons between Germany, the United States, and Korea, and talks about support from the economic policies of the states. INDUSTRY 4.0 is also an important growth factor in China. Li [15] notes not only the technological but also sociological context of these changes, and finds an upward trajectory in China in manufacturing capability development, research and development commitment, and human capital investment. It is very important to see the connections between R&D, INDUSTRY 4.0, the development of human capital, and economic political strategies of individual states. No less important is the monitoring of the links between science and research, innovation, INDUSTRY 4.0, and the development of the workforce to meet these strategies.

Min et al. [16] focuses on a very broad innovation framework in the US and the smartization strategy in Japan. The authors show that developed countries are pushing nation-wide innovation strategies. Similarly, China is pursuing the Made in China 2025, and Korea has announced the Manufacturing Industry Innovation 3.0 strategy. Min et al. [16] provides a comparative study on industrial spillover effects among Korea, China, the USA, Germany, and Japan, especially the spillover effect of the Information and Communication Technology (ICT) industry and equipment (the foundation of smart manufacturing through convergence with the ICT industry). Practical implications of their findings are that Germany needs to revitalize the ICT industry to strengthen its manufacturing industry. It is a very important finding with many implications for the Czech Republic, because Czech industry is an important subcontractor to the German industry. The connection of Czech companies to German industry has been the basis for the growth of the Czech GDP in the last 25 years. There is transfer know-how in subcontracting chains, which enhances the development of workforce skills. At the same time, there is a very strong pressure to increase production efficiency and reduce costs, which is manifested by the pressure to increase labor productivity, the introduction of robotics, and the successful implementation of INDUSTRY 4.0 in the Czech Republic.

Macroeconomics captures economic growth through production functions, where the basic variables are human resources and capital, taking into account that capital is financial, and also includes the impact of new technologies and processes, including INDUSTRY 4.0. However, it would be a fundamental mistake to separate it from human resources, in their quantitative and, above all, qualitative assessment (qualifications, personalities responsive to change, teamwork, etc.). Most of the theoretical work, however, does not take into account the required growth of labor force qualification for INDUSTRY 4.0. For example, Grassetti and Hunanyan [17] used a neoclassical one-sector growth model with differential savings, while assuming a Kadiyala production function that shows a variable elasticity of substitution symmetric with respect to capital and labor. Authors declare that thanks to the proposed methodology, the government can select a proper economic policy to reduce production costs without decreasing the capitalization trend of the economy. The theory, however, does not correspond to the issue of investment in education and the growth of labor force skills needed to grow productivity and the whole economy. The causes of economic crises are perceived as external factors of the growth model.

Some authors are looking for the causes of cyclical macroeconomic behavior. Devezas and Corredine [18] focused on the effective causality of long-term macroeconomic rhythms, commonly referred to as long waves or Kondratieff waves. The authors have demonstrated that the unfolding

of each structural cycle of a long wave is controlled by two parameters: the diffusion-learning rate $\delta$ and the aggregate effective generation $tG$, whose product is maintained in the interval $3 < \delta tG < 4$ (deterministic chaos) of social systems. For the development of INDUSTRY 4.0, it is particularly important to define the basic variables that the authors have included in their model and which show the importance of the human factor for long-term development. Dosi et al.'s [19] model results show that seemingly more rigid labor markets and labor relations are conducive to coordinating successes, leading to higher and smoother growth. Emphasis on the development and organization of the workforce can be crucial for the development of INDUSTRY 4.0.

On the other hand, for businesses that are using INDUSTRY 4.0 technologies and practices, it is important to anticipate the near-term (not long-term) macroeconomic development of the economy. For Czech companies, it is mainly macroeconomic developments in Germany, but also in many other European countries and the world. The anticipation of close macroeconomic developments is the responsibility of a number of institutions. For a short time horizon (typically several months), the statistical authorities perform standardized statistical surveys—purchasing managers' expectations, but also IFO Index, ZEW indicators, and more. Homolka and Pavelková [20] examined the predictive power of the ZEW sentiment indicator in the case of the German automotive industry, which is crucial for the Czech economy. The ZEW Indicator of Economic Sentiment is a leading indicator of the German economy, similar to the IFO index. All of these indicators can reduce the uncertainty of businesses that want to invest in R&D and INDUSTRY 4.0, but fear that their investment can be destroyed by a future economic crisis.

INDUSTRY 4.0 is a very new phenomenon that does not go beyond this decade. Macroeconomic research of this phenomenon, therefore, faces insufficient methodological definition and coverage of data by international and national statistical institutions. The definition of Research and Development (R&D) is much older and more sophisticated, as developed by the Organization for Economic Cooperation and Development OECD in 1963 in its first Frascati Manual, which started the process of institutional R&D surveys. Nowadays, R&D surveys regularly monitor, among other things, expenditure on ICT equipment, software, biotechnology, and nanotechnology, funded from both private and public sources. Subsequently, there has been an effort to link the R&D methodological system with the System of National Accounts (SNA), which has been methodically managed by the United Nations (UN) since 1947. In the latest version of SNA 2008, both systems were already compatible and usable for the analyses of Gross Domestic Product (GDP). SNA and GDP are important for analyzing periods of recession or boom periods. Some authors (e.g., Monsori [21]) see a very close relationship between INDUSTRY 4.0 and R&D. In this article, the concept of R&D, which is more methodically covered, has been used to research the new INDUSTRY 4.0 phenomenon. However, it would certainly be worth exploring the characteristics and degree of overlap of the two important concepts in the future.

## 3. Materials and Methods

Research Question 1 (RQ1): Is the number of economic subjects investing in R&D in the Czech Republic in times of economic crisis (recession) decreasing?

Research Question 2 (RQ2): Do the economic sectors of the Czech Republic (business, government, university, and non-governmental non-profit sectors) develop equally, or do they each have specific conditions and are affected by specific factors?

Research question 3 (RQ3): Can human resources be a limiting factor for the development of R&D and INDUSTRY 4.0?

Data: Data on subjects investing in R&D results, including INDUSTRY 4.0 technologies and methods, were taken from the Czech Statistical Office [22–26]. Supporting the introduction of R&D results into production and business processes is part of the Czech Republic's economic policy strategy. Companies in the Czech Republic interested in obtaining scientific grants and subsidies from the economic policy of the state are voluntarily registered with CZ-NACE 72 Research and Development (R&D in the technical industry has a number CZ-NACE 72.19.2). The Czech Classification of Economic

Activities CZ-NACE is derived from the International Standard Industrial Classification of All Economic Activities (ISIC). The Czech Statistical Office sends to these companies the Annual Report on R&D VTR 5-01 [24–26]. It is the collection of internationally comparable data on human and financial resources in the field of research and development in the Czech Republic. This collection of statistics is in line with Decision No. 1608/2003/EC of the European Parliament and of the Council on a methodology for statistics on science and technology for the European Union, within the Europe 2020 Strategy. The collection monitors all R&D subjects (CZ-NACE 72), regardless of whether R&D is their main or secondary economic activity. The Czech Statistical Office uses the method of combining exhaustive and sample surveys [24–26].

Method: The method of multivariate statistics with dummy variables was used to analyze the impact of economic contractions on the number of R&D subjects. Multiple regression is an extension of the bivariate linear regression firms [27], predicting a variable from another's scores. Dummy variables (nominal variables coded 1 = "recession", 0 = "non-recession") were used for the purpose of identifying the influence of the economic cycle-phase recession.

Procedure: Firstly, Model 1 was created and tested; the model included R&D entities from all four sectors of the Czech economy surveyed (business, governmental, university, and non-profit sectors). Secondly, different conditions (especially legal and economic) and different factors of influence (in particular taxes, grants and subsidies) in individual sectors were expected. Therefore, Model 2 was subsequently created and this model examined each sector separately. The results of both models were then compared and discussed. Subsequently, data on the development of human resources in R&D in the Czech Republic and their possible impact on the development of INDUSTRY 4.0 were presented and discussed.

## 4. Results

The results are presented in connection with the research questions (RQ1–RQ3).

*4.1. Research Questions 1 and 2*

4.1.1. Recession and Non-Recession Model 1—All Sectors Together

Table 1 describes the number of Czech subjects reporting R&D results over the years 2007 to 2017. The recession periods are based on the definition of declining GDP and are marked. The economic crises in 2009 and 2012–2013 were accompanied not only by a decline in GDP, but also by a sharp decline in the production performance of CZ-NACE C Manufacturing in the Czech Republic (The Czech Classification of Economic Activities CZ-NACE is derived from the International Standard Industrial Classification of All Economic Activities).

**Table 1.** Numbers of Research and Development (R&D) subjects in the Czech Republic in sectors 2007–2017 [24–26].

| Sector | 2007 | 2008 | 2009 | 2010 | 2011 | 2012 | 2013 | 2014 | 2015 | 2016 | 2017 |
|---|---|---|---|---|---|---|---|---|---|---|---|
| Business enterprise | 1736 | 1766 | 1872 | 2107 | 2237 | 2312 | 2299 | 2368 | 2387 | 2355 | 2628 |
| Government | 224 | 222 | 223 | 219 | 209 | 195 | 199 | 195 | 196 | 195 | 199 |
| Higher education | 186 | 185 | 187 | 193 | 202 | 203 | 208 | 213 | 228 | 227 | 229 |
| Non-gov-non-profit | 58 | 60 | 63 | 68 | 72 | 68 | 62 | 64 | 59 | 53 | 58 |
| GDP [1] real | 3964 | 4070 | 3874 [2] | 3962 | 4033 | 4001 [2] | 3981 [2] | 4089 | 4307 | 4412 | 4601 |

[1] Database of National Accounts. Czech Statistical Office (CZSO) in billion Czech crowns (CZK) [23]. [2] Recessions (decline in GDP).

The number of subjects of the Czech Republic declaring the results of R&D in the years 2007 to 2017 raises from 2021 to 3114 subjects [24–26]. The total number of all entities in the Czech Republic increased from 570,000 to 668,000 subjects in the same period (the non-profit sector consisted of

120–150,000 subjects, the government sector had 18,000 subjects, and the business sector had developed from 409,000 to 502,000 subjects) [23].

Table 2 shows descriptive statistics of all four sectors of the Czech economy in terms of the number of entities that declare R&D results.

**Table 2.** Descriptive statistics of each sector.

| Sector | n | Mean | Sd | Median | Min | Max | Se |
|---|---|---|---|---|---|---|---|
| Business enterprise s. | 11 | 2187.91 | 285.03 | 2299 | 1736 | 2628 | 85.94 |
| Government s. | 11 | 206.91 | 12.64 | 199 | 195 | 224 | 3.81 |
| Higher education s. | 11 | 205.55 | 17.03 | 203 | 185 | 229 | 5.14 |
| Non-gov. non-profit s. | 11 | 62.27 | 5.5 | 62 | 53 | 72 | 1.66 |

As can be seen for business enterprise sector the numbers of subject is much higher than for other sectors. This is the main reason why we use the logarithm function in our first model (Model 1) to eliminate the different levels of measurement. The fact that government and higher education sectors have very similar mean and median number of subjects is also interesting.

The following overview shows the results of the bivariate linear regression forecasting variable from other scores. Dummy variables (nominal variables coded 1 = "recession", 0 = "non-recession") were used for the purpose of identifying the influence of economic cycle phase recession.

Model 1 is given in Equation (1):

$$\log(\#subjects) = \beta_{0,j} + \beta_{1,j}(t) + \beta_{2,j}(recession) + \epsilon_{i,j}, \tag{1}$$

where #subjects denote the number of subjects, j is the index of each sector.

The logarithm of the number of R&D entities working in all sectors (business, government, higher education, and non-profit) is determined by the year and whether or not there has been a recession.

Results of Model 1 (all sectors together) are presented in Table 3.

**Table 3.** Results of linear mixed-effect Model 1.

| | Value | Std. Error | DF | t-Value | p-Value |
|---|---|---|---|---|---|
| intercept | 5.557 | 0.6861 | 38 | 8.09836 | 0.0000 |
| t | 0.00954 | 0.0131 | 38 | 0.7309 | 0.4693 |
| recession | 0.00585 | 0.0184 | 38 | 0.318373 | 0.7519 |

Table 3 shows results of linear mixed effect model. As can be seen the only statistically significant variable is the intercept with the value of 5.557. When the p-value for t and recession variable are greater than 0.05 we can conclude that these two variables are not statistically significant. Interestingly, the t variable was observed to be insignificant, which can be firstly seen as counterintuitive. However, from Table 1 we can see that the number of subjects rises only for two sectors (Business enterprise sector and Higher education sector) over time, as can be seen the number of subjects in the government sector decreases. This is the main reason why the time variable is insignificant if we analyze all four sectors together.

4.1.2. Recession and Non-Recession Model 2—Each Sector Separately

Model 2 works with the same data as Model 1, but it examines each sector of the Czech economy separately. Therefore, it is no longer necessary to use the logarithm function.

Model 2 (separate results for each sector) is given in Equation (2):

$$\#subjects = \beta_0 + \beta_1(t) + \beta_2(recession) + \epsilon_i, \tag{2}$$

where #subjects are number of subjects.

Results of Model 2 (Business enterprise sector) are presented in Table 4.

**Table 4.** Model 2 results for business enterprise sector.

| Predictors | Estimates | CI | p |
|------------|-----------|-----|---|
| (Intercept) | 1685.42 | 1547.18–1823.67 | <0.001 |
| t | 82.01 | 62.95–101.08 | <0.001 |
| Recession | 38.18 | −97.19–173.55 | 0.596 |
| Observations | 11 | | |
| $R^2$/adjusted $R^2$ | 0.899/0.874 | | |

The results of linear regression (Table 4) show that the p-value is lower than 0.05 for intercept and time variable. As can be seen, each year the number of subjects in this sector rises by a value of 82. It also can be seen that the dummy variable (recession) is insignificant. The $R^2$ of this model is 0.89, which can be interpreted as this model explaining 89.9% of variation.

Results of Model 2 (Non-government non-profit sector) are presented in Table 5.

**Table 5.** Model 2 results for non-governmental non-profit sector.

| Predictors | Estimates | CI | p |
|------------|-----------|-----|---|
| (Intercept) | 64.52 | 56.74–72.30 | <0.001 |
| t | −0.48 | −1.56–0.59 | 0.403 |
| Recession | 2.39 | −5.23–10.01 | 0.556 |
| Observations | 11 | | |
| $R^2$/adjusted $R^2$ | 0.142/−0.073 | | |

The results of linear regression (Table 5) show that the p-value is lower than 0.05 for the intercept only. This model has very poor explanatory power. It can be seen from the data that this sector has a very subtle number of subject fluctuations, which can be explained by the intercept itself. The result shows that the number of subjects can be predicted by intercept, which is 64 subjects with no growth or descent annually.

Results of Model 2 (Government sector) are presented in Table 6.

**Table 6.** Model 2 results for Government sector.

| Predictors | Estimates | CI | p |
|------------|-----------|-----|---|
| (Intercept) | 229.07 | 220.85–237.29 | <0.001 |
| t | −3.47 | −4.60–−2.34 | <0.001 |
| Recession | −4.89 | −12.94–3.16 | 0.268 |
| Observations | 11 | | |
| $R^2$/adjusted $R^2$ | 0.819/0.774 | | |

The results of linear regression (Table 6) show that the p-value is lower than 0.05 for the intercept and time variable. As can be seen, each year the number of subjects in this sector decreases by a value of 3. It also can be seen that the dummy variable (recession) is insignificant. The $R^2$ of this model is 0.82, which can be interpreted as this model explaining 82% of variation.

Results of Model 2 (Higher education sector) are presented in Table 7.

The results of linear regression (Table 7) show that the p-value is lower than 0.05 for the intercept and time variable. As can be seen each year, the number of subjects in this sector rises by a value of 5. It also can be seen that the dummy variable (recession) is insignificant. The R^2 of this model is 0.96, which can be interpreted that this model explains 96% of variation.

**Table 7.** Model 2 results for higher education sector.

| Predictors | Estimates | CI | p |
|---|---|---|---|
| (Intercept) | 176.97 | 172.23–181.71 | <0.001 |
| t | 4.95 | 4.29–5.60 | <0.001 |
| Recession | −4.01 | −8.65–0.63 | 0.129 |
| Observations | 11 | | |
| $R^2$/adjusted $R^2$ | 0.967/0.959 | | |

## 4.2. Research Questions 3

### 4.2.1. Human Resources Working in R&D in the Czech Republic

The focus of this research is also on human resources for R&D and INDUSTRY 4.0 projects in relation to the period of recession or non-recession. Table 8 shows an overview of the development of R&D personnel by sector and field of science. Recession periods are highlighted.

**Table 8.** R&D personnel by sector of the Czech Republic by fields of science 2007–2017 [24–26].

| Sector | 2007 | 2008 | 2009 [1] | 2010 | 2011 | 2012 [1] | 2013 [1] | 2014 | 2015 | 2016 | 2017 |
|---|---|---|---|---|---|---|---|---|---|---|---|
| **Business enterprise** | 30,640 | 31,660 | 32,375 | 34,658 | 37,437 | 41,445 | 44,255 | 48,194 | 49,252 | 51,069 | 55,232 |
| Science | 4513 | 4226 | 4675 | 5872 | 7156 | 8494 | 9810 | 11,558 | 11,812 | 12,148 | 13,361 |
| Technical sciences | 23,830 | 25,114 | 25,279 | 26,063 | 27,085 | 29,079 | 31,614 | 32,673 | 33,697 | 34,262 | 37,727 |
| Medical sciences | 744 | 801 | 820 | 1196 | 1174 | 1156 | 993 | 1022 | 1080 | 1212 | 1168 |
| Agricultural sciences | 1256 | 1243 | 1230 | 1255 | 1305 | 1622 | 1151 | 1262 | 1302 | 1302 | 1131 |
| Social Sciences | 271 | 242 | 343 | 246 | 686 | 1077 | 666 | 1666 | 1347 | 2116 | 1818 |
| Humanities | 26 | 35 | 29 | 27 | 29 | 16 | 21 | 14 | 14 | 28 | 27 |
| **Government** | 15,470 | 15,559 | 15,402 | 15,029 | 15,313 | 15,482 | 15,996 | 16,177 | 16,705 | 16,615 | 17,941 |
| Science | 8276 | 8544 | 8586 | 8532 | 8576 | 9165 | 9464 | 9544 | 9517 | 9686 | 10,094 |
| Technical sciences | 464 | 557 | 484 | 413 | 414 | 411 | 488 | 492 | 559 | 554 | 604 |
| Medical sciences | 2195 | 2124 | 2168 | 1948 | 2000 | 1931 | 1935 | 1802 | 1980 | 1605 | 2029 |
| Agricultural sciences | 1239 | 1136 | 956 | 1089 | 1107 | 829 | 842 | 993 | 1225 | 1211 | 1319 |
| Social Sciences | 1029 | 893 | 820 | 777 | 874 | 874 | 758 | 785 | 943 | 973 | 1112 |
| Humanities | 2267 | 2305 | 2388 | 2270 | 2342 | 2272 | 2509 | 2562 | 2481 | 2586 | 2783 |
| **Higher education** | 26,735 | 26,993 | 27,694 | 27,844 | 29,149 | 30,301 | 32,173 | 32,680 | 33,891 | 31,915 | 34,234 |
| Science | 2887 | 3431 | 3286 | 3788 | 5818 | 5864 | 6512 | 6804 | 6743 | 6528 | 7517 |
| Technical sciences | 8573 | 8463 | 8492 | 8304 | 7436 | 8947 | 8308 | 8955 | 9530 | 9118 | 9383 |
| Medical sciences | 6347 | 6439 | 7386 | 6648 | 6773 | 6172 | 7340 | 7149 | 7361 | 6677 | 7213 |
| Agricultural sciences | 2353 | 2356 | 2657 | 2538 | 2003 | 2166 | 2276 | 2097 | 1927 | 1861 | 1996 |
| Social Sciences | 4574 | 4367 | 3393 | 3487 | 4959 | 4597 | 5447 | 5365 | 5550 | 5194 | 5509 |
| Humanities | 2001 | 1937 | 2480 | 3079 | 2160 | 2555 | 2290 | 2310 | 2780 | 2537 | 2616 |
| **Non-gov. non-profit s.** | 236 | 296 | 317 | 372 | 384 | 300 | 290 | 302 | 280 | 276 | 327 |
| Science | 26 | 29 | 74 | 103 | 84 | 61 | 55 | 73 | 79 | 62 | 68 |
| Technical sciences | 69 | 93 | 28 | 49 | 56 | 47 | 40 | 51 | 23 | 36 | 59 |
| Medical sciences | 5 | 16 | 11 | 6 | 23 | 10 | 3 | 1 | - | - | 8 |
| Agricultural sciences | 13 | 25 | 24 | 16 | 12 | 8 | 4 | 10 | 4 | 5 | 5 |
| Social Sciences | 115 | 106 | 156 | 146 | 174 | 155 | 183 | 166 | 174 | 165 | 179 |
| Humanities | 7 | 27 | 24 | 52 | 36 | 19 | 5 | 1 | 1 | 8 | 8 |

[1] Recessions.

The business enterprise sector has a very strong predominance of R&D personnel in the field of technical sciences. This is fundamentally different from other sectors, specifically the university sector, which has a high share of technical scientists, but a high proportion of workers are also seen in the fields of natural sciences and medicine. This comparison shows the strong unilateral orientation of CZ-NACE C Manufacturing in the business sector, which also directs the current INDUSTRY 4.0 projects in the Czech Republic.

Figure 1 shows the international comparison of the number of R&D personnel in selected countries of the world during 2005–2015 [24]. Data on human resources working in R&D in the United States was not available.

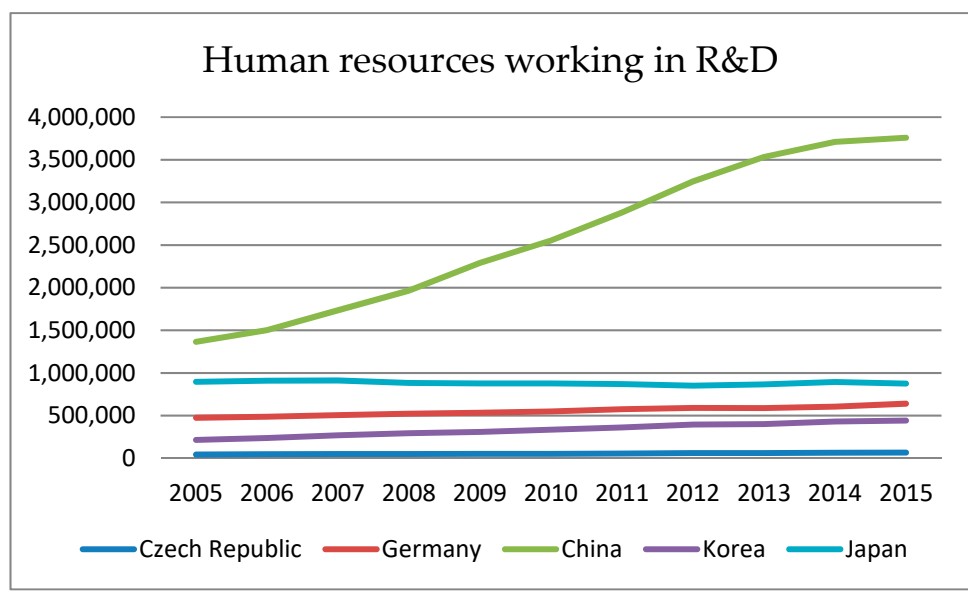

**Figure 1.** Human resources working in R&D—international comparison for the period 2005–2015 [24].

Figure 1 shows an international comparison of R&D personnel development numbers in the Czech Republic and Germany, showing that numbers are increasing, which gives good conditions for the development of INDUSTRY 4.0 projects. Compared with China, however, the growth rate is relatively slow, which may worsen future competitiveness.

4.2.2. Future Human Resources for R&D in the Czech Republic

The future competitiveness of the Czech Republic will be influenced mainly by the number of graduates of higher education institutions working in the area of R&D, especially in the field of technical sciences. Table 9 shows current vacancies for graduates.

**Table 9.** The statistics of vacancies for university graduates and high school graduates in the Czech Republic on the date 18 January 2019 [28].

| Field | Total Number of Vacancies for University [1] Graduates | Total Number of Vacancies for School [2] Graduates |
|---|---|---|
| Administration | 67 | 2150 |
| Transport | 9 | 994 |
| Finance | 38 | 1352 |
| Information Technology | 347 | 854 |
| Culture and Sport | 23 | 290 |
| Management | 16 | 357 |
| Trade and Tourism | 65 | 4553 |
| Defense and Security | 3 | 720 |
| Services | 0 | 1351 |
| Building Industry | 44 | 1634 |
| Science and Research | 144 | 98 |
| Education | 378 | 351 |
| Operations | 256 | 10,131 |
| Health Care | 1112 | 1912 |
| Agriculture and Forest | 26 | 239 |
| Law | 44 | 8 |

[1] University (college) graduates. [2] Jobs for applicants with completed secondary education.

Table 9 shows current vacancies for university graduates, where the R&D area in the Czech Republic is not large. For the further development of INDUSTRY 4.0, these vacancies will need to be expanded and more attractive to young people.

## 5. Discussion and Conclusions

INDUSTRY 4.0 is a very important topic that can be explored from a variety of perspectives. In the introduction to this article at least some of them have been highlighted. The research has revealed a number of definitions of INDUSTRY 4.0, but the authors have not found a precise definition of the relationship of INDUSTRY 4.0 to another important topic, namely the development of science and research. The authors of this article perceive a close relationship between INDUSTRY 4.0 and R&D, but this relationship itself is not the subject of research. The authors see it as important to seek a more precise definition of this relationship in future research.

A very interesting topic is the process of implementing INDUSTRY 4.0 (or R&D projects) in relation to the macroeconomic cycle stage. The arrival of an economic recession especially affects the introduction of new projects in companies, as Pavelková has stated [4]. Homolka and Pavelková [20] examined the predictive power of the ZEW sentiment indicator in the case of the German automotive industry, which is important for the Czech economy. Leading macroeconomic indicators can reduce the uncertainty of businesses that want to invest in R&D and INDUSTRY 4.0, but there is still fear that their investment could be destroyed by an economic crisis. The authors of this article based their work on these assumptions and focused on the question of whether the evolution of the business cycle measured by GDP at constant prices affects the numbers and selected indicators of R&D results, both in the business sector and in the government, universities, and non-governmental non-profit organization sectors.

The research was divided into three research questions, of which the first two research questions examined the number of R&D entities in the Czech Republic during 2007–2017. The research horizon included two periods of economic recession, in 2009 and during 2012–2013, but the research is not robust enough to be considered representative. It is possible to discuss whether R&D entities voluntarily register with CZ NACE 72 due to their research activities or due to subsidies and grants. On the other hand, the methodology used for monitoring R&D subjects is internationally recognized and widely implemented.

The method of multivariate statistics with dummy variables was used to analyze the impact of economic recession on the number of R&D subjects. The results (Table 3) show that the recession variable in Model 1 is insignificant. This can have at least three possible explanations. The first explanation is that recession does not have any explanatory power. The second explanation is that recessions are relatively very rare events and there is not enough data to show their explanatory power. The third explanation is that each sector (business, governmental, university, and non-profit sectors) has different conditions (especially legal and economic) and different factors of influence (in particular taxes, grants, and subsidies). This idea is supported by the fact that almost in every year (no matter if recession strikes or not) the number of subjects rising or descending is continuous. Therefore, Model 2 was subsequently created, and this model examined each sector separately.

Results for Model 2 are presented separately from each sector (Tables 4–7). The results of linear regression for the business sector (Table 4) show that the p-value is lower than 0.05 for the intercept and the time variable. As can be seen, each year the number of subjects in this sector rises by a value of 82. It also can be seen that the dummy variable (recession) is insignificant. The $R^2$ of this model is 0.89, which can be interpreted that this model explains 89.9% of variation. The results of linear regression show that the p-value is lower than 0.05 for intercept only. The result of the linear regression for the non-governmental non-profit sector (Table 5) has very poor explanatory power. This sector has very subtle number of subject fluctuations, which can be explained by the intercept itself. The result shows that the number of subjects can be predicted by the intercept, which is 64 subjects, with no growth or descent annually. The results for the governmental sector (Table 6) show that the number of subjects in

this sector decreases by a value of 3 each year. It can also be seen that the dummy variable (recession) is insignificant. The $R^2$ of this model is 0.82 which can be interpreted as this model explaining 82% of the variation. This result shows that the government sector is not affected either by economic recession or by grants and subsidies, and is steadily declining. This is surprising and alarming. On the contrary the results of the higher education sector (Table 7) show that the p-value is lower than 0.05 for the intercept and time variable. As can be seen, each year the number of subjects in this sector rises by a value of 5. Although a significant number of the universities in the Czech Republic are public, it is positive that the number of R&D subjects is steadily increasing. Here you can see the positive impact of grants, especially from European Union funds. It also can be seen that the dummy variable (recession) is insignificant. The $R^2$ of this model is 0.96, which can be interpreted as this model explaining 96% of the variation.

The results of both models show that the impact of recession (dummy variable) is not significant. The growth of the number of R&D entities in the CR is much more influenced by other factors (mainly tax relief and grants supporting R&D from EU funds).

Subsequently, data on the development of human resources in R&D in the Czech Republic and their possible impact on the development of INDUSTRY 4.0 (Research question 3) were presented and discussed. Table 8 presents an overview of the development of R&D personnel by sector and field of science. The business enterprise sector in the Czech Republic has a very strong predominance of R&D personnel in the field of technical sciences. This is fundamentally different from other sectors. Specific is the university sector, which has a high share of technical scientists, but a high proportion of workers are also seen in the fields of natural sciences and medicine. This comparison shows the strong unilateral orientation of Manufacturing in the business sector, which also directs the current INDUSTRY 4.0 projects in the Czech Republic. Strong pressure on increasing the number of R&D personnel, however, is not reflected in the currently offered vacancies for university graduates (Table 9). For the further development of INDUSTRY 4.0 projects in the Czech Republic, there is a need to increase and make attractive the qualified job positions that are important for these projects.

Min et al. [16] emphasizes that Germany needs to revitalize the ICT industry to strengthen its manufacturing industry. The Czech industry is an important subcontractor for the German industry. The development of INDUSTRY 4.0 projects in the Czech Republic could boost the competitiveness of both the Czech Republic and Germany, but it is conditioned by a change in the structure of the supply of qualified positions in the necessary structure and relevant demand from university graduates.

The research findings of this article also showed that current EU grant support and tax relief appear to be more important factors for the development of new R&D projects in the Czech Republic than the effects of the economic recession. In terms of longer time horizons, however, the effects of the business cycle cannot be ignored, and their impact on the future development of INDUSTRY 4.0 may be strengthened over time.

## 6. Conclusions

The current INDUSTRY 4.0 projects in the Czech Republic bring a number of cutting-edge solutions that can be used globally. There are also research institutions dealing with advanced manufacturing in cooperation with companies, which is a step closer towards smart factories. A few of these are listed below.

- The Research Center of Manufacturing Technologies (RCMT) at the Czech Technical University cooperates with industry, especially on topics of advanced simulation models, virtual prototyping and virtual testing, development of advanced feed drive control techniques and vibration suppression methods, advanced monitoring and diagnostics of machine tool condition, multi-axis machining technology, etc. Introducing R&D results helps the industry address its challenges. Machine tools can use, for example, a model to predict micromilining cutting forces, which estimates the tool's deflection and the real tool-path during the micromilling process [29]. Manufacturing also needs methods to monitor tool wear; for instance, one of such methods uses

> the application of a ceramic piezoelectric sensor mounted on the tool holder in the turning machine to monitor vibration signals due to the flank wear progression [30]. A very important objective, not only in the automotive industry, is to achieve a good surface quality directly from machining without any additional manual work, especially with use of advanced high-strength steels (AHSS). For instance, López de Lacalle's technological model of the milling process estimates values of cutting forces and can offer manufacturers a reduction of production and lead times [31].

- Many companies have their own research facilities, for instance the company Prusa Research focuses on manufacturing of 3D printers and is a global leader in its category thanks to the innovation of using a full metal nozzle and PCB heated bed [32]. Smart factories are in the process of automation and robotization of their production.

- Hybrid manufacturing technologies are used, for instance, the technology developed by Kovosvit and RCMT, which enables manufacturing with additive technology and welding of various combinations of materials, welding of functional surfaces, parts, and details, repairs, creation of full parts with internal channels, shell parts, and hollow parts, all in combination with machining. The rate of growth of parts of different steels is in the range of 0.2 to 1.0 kg/h [32].

- The remote diagnostic is useful and popular in smart factories, for instance, Wikov launched the remote diagnostic tool for online monitoring of gearbox condition. This is a system for the complete driveline and enables optimization of the maintenance plan for maximum availability and minimum downtime. Various data, such as vibration, temperature, speed, pressure, and other parameters are monitored and postprocessed. Outputs are accessible in real time via a web-based interface. The software's advanced algorithms can detect gear-teeth and bearing damages at a very early stage, and thus prevent major gearbox damage [32].

There are many fields in which IoT and monitoring or high-tech approaches are in use. Multiple IoT networks are already being constructed and are ready for use in the country.

The results of both models show that the impact of economic recession on the R&D entities in the CR is not significant nowadays. These entities in Europe are much more influenced by other factors, for instance, grants to support R&D from EU funds, such as Horizon 2020 [32]. The number of R&D subjects in the Czech Republic grew in some sectors (business enterprise sector) in the monitored period of two economic recessions. Usually, in a period of economic recession, the decline in economic performance is accompanied by a decline in the number of economically active entities. From this perspective, it is possible to supplement the scientific literature with findings from this paper.

The added value of Models 1 and 2 is seen primarily in the business sector, which is sufficiently large and dynamically evolving. The dummy variable is not significant in model 2 (Table 4, business enterprise sector), so it can be said that the number of R&D subjects in the business sector is not dependent on the recession. Other sectors are small in number of entities, for example, non-profit non-governmental sector, which has about 150,000 subjects, but only about 60 entities declare R&D results. The study wanted to show the difference in business sector development from other sectors. The ability of the business sector to develop R&D projects, even in times of economic recession, could be used in economic policy. The effective direction of grants in this area could then accelerate the faster return of the economy to economic growth.

This study has its limitations, mainly in the area of available data, because the recession is a rare phenomenon and we have only three data points for each sector (data on the number of R&D entities are collected only once a year; Report VTR 5-01). On the other hand, this phenomenon is very interesting from perspectives of both scientific literature and business, so it would be beneficial to develop this research and perform calculations on a representative set of data.

The other limitation of this research is that in this article, the concept of R&D, which is more methodically covered, has been used to research the new INDUSTRY 4.0 phenomenon. However, it would certainly be worth exploring the characteristics and degree of overlap of the two important concepts.

**Author Contributions:** Conceptualization, J.Š. and T.U.; methodology, J.Š.; software, T.U.; validation, L.P., and E.S.; formal analysis, T.U.; investigation, J.Š.; resources, L.P.; data curation, J.Š. and E.S.; writing—original draft preparation, J.Š.; writing—review and editing, L.P. and E.S.; visualization, T.U.; supervision, J.Š.; project administration, L.P.; funding acquisition, L.P.

**Funding:** This research was funded by the Internal Grant Agency of FaME TBU IGA/FaME/2018/001 "Leading indicators in the buying behavior of companies in B2B markets" and RO/2018/08 "Research on qualitative and quantitative changes in demand on the Czech labor market with the introduction of INDUSTRY 4.0".

**Acknowledgments:** The authors wish to thank the Internal Grant Agency of FaME TBU IGA/FaME/2018/001 "Leading indicators in the buying behavior of companies in B2B markets" and RO/2018/08 "Research on qualitative and quantitative changes in demand on the Czech labor market with the introduction of INDUSTRY 4.0".

**Conflicts of Interest:** The authors declare no conflict of interest.

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
