# Peer review of "Implementation of R&D Results and Industry 4.0 Influenced by Selected Macroeconomic Indicators"

_applsci, doi:10.3390/app9091846_

Round 1
Reviewer 1 Report
The paper titled “Implementation of R&D results and industry 4.0 influenced by selected macroeconomic indicators” aims at looking for macroeconomic links between GDP growth and development of research and science in the Czech Republic.
Even though the paper is interesting, I feel that it is publishable subject to some, rather, major, revision. The most important remark is that the authors need to highlight the contribution (i.e. the added value) of the paper to the scientific literature as well as the lessons that can be drawn from the paper’s findings and the corresponding policy suggestions that can be made
I hope that the following specific comments may help the authors to revise their paper.
The introductory section is too lengthy, and the aim of the paper is stated too late. Definitely, this section needs some rearrangement.
In the introductory section, the authors may provide an explanation why this study is (can be) interesting for an international audience. Is the Czech Republic a “representative” case-study?
A literature review section is missing and must be added. Some parts of the introductory section may divert to this literature review section. It is not clear what is the theoretical framework that underpins the empirical analysis of the paper.
Table 1: It would be useful to know the total number of subjects (i.e. R&D and non-R&D) in the Czech Republic.
Figure 1 is not necessary.
I cannot understand the added value of Models 1 and 2 given that the majority of the independent variables are statistically non-significant.
Figure 2: Why the Czech Republic is compared to Germany, China, Korea and Japan?
To my point of view, conclusions are not linked adequately with the empirical analysis.
Author Response
Thank you very much for your valuable comments and suggestions - we use all of them to improve our paper.
Point 1:.Even though the paper is interesting, I feel that it is publishable subject to some, rather, major, revision. The most important remark is that the authors need to highlight some contribution (i.e. the added value) of the paper to the scientific literature as well as the lesson that can be drawn from the paper´s findings and corresponding policy suggestions that can be made.
Response 1: We made important changes, especially in the Introduction section and in the Conclusion section. We have highlighted the added value of the contribution and the corresponding macroeconomic policy proposals in the Conclusion section.
Point 2:.The introductory section is too lengthy, and the aim of the paper is stated too late. Definitely, this section needs some rearrangement.
Response 2: We have shortened and structured the introduction according to your recommendations and separated it from the literature review.
Point 3:.In the Introductory section, the authors may provide an explanation why this study is (can be) interesting for international audience. Is the Czech Republic a “representative” case-study?
Response 3: We added a new paragraph in the Introduction section to explain the international context and impacts especially for Europe and all partners who cooperate with Europe countries.
Point 4:.A literature review section is missing and must be added. Some parts of the introductory section may divert into this literature review section. It is not clear what is the theoretical framework that underpins the empirical analysis of the paper.
Response 4: We added the Literature review section and highlighted the theoretical framework.
Point 5:.It would be useful to know the total number of subjects (i. e. R & D and non- R & D) in the Czech Republic.
Response 5: Table 1 comments were supplemented by the total number of subjects in the Czech Republic between 2007 and 2017, including sector breakdown.
Point 6:.Figure 1 is not necessary.
Response 6: Figure 1 has been canceled. Figure 2 was renumbered in Figure 1.
Point 7:.I cannot understand that added value of Models 1 and 2 given that the majority of the independent variables are statistically non-significant.
Response 7: Most independent variables are not significant because the recession is a rare phenomenon and we have only three data points for the sector (data on the number of R&D entities are collected only once a year: Report VTR 5-01). On the other hand, this phenomenon is very interesting for both scientific literature and business, so we decided to publish the results even under these circumstances.
Point 8: Figure 2: Why the Czech Republic is compared to Germany, China, Korea and Japan?
Response 8: Germany is the Czech most important business partner (40% of the Czech exports) and China, Korea and Japan are very important business partners as well (they have numerous direct investments in the Czech Republic, especially into the automotive industry).
Reviewer 2 Report
The paper talks about the relation of INDUSTRY 4.0 and research and development (R & D) projects. This article explores the impact of the two economic recessions in 2009 and 2012-2013 on the number of R & D entities and human resources involved in R & D in the Czech Republic. The method of multivariate statistics with dummy variables was used. The authors indicate that current EU grant support, tax relief and other specific factors appear to be more important for the development of R & D projects in the Czech Republic than the effects of economic recession. However, I have some comments which I would like to be addressed before the acceptance of the paper.
Major Comments:
1. I recommend the authors to follow the MDPI’s proper Introduction format. Currently, Introduction is too long and it distract the reader’s attention. The introduction should briefly place the study in a broad context and highlight why it is important. It should define the purpose of the work and its significance, including specific hypotheses being tested. Finally, briefly mention the main aim of the work and highlight the main conclusions. Keep the introduction comprehensible to scientists working outside the topic of the paper.
2. I suggest authors to put a new section after introduction that only explains the relationship between INDUSTRY 4.0 and research and development (R & D) projects.
3. Briefly describe the limitations at the end of conclusion section.
Author Response
Thank you very much for your helpful comments and suggestions - we use all of them to improve our paper.
Point 1: I recommend the authors to follow the MDPI´s proper Introduction format. Currently, Introduction is too long and it distracts the reader´s attention. The introduction should briefly place the study in a broad context and highlight why it is important. It should define the purpose of the work and its significance, including specific hypotheses being tested. Finally, briefly mention the main aim of the work and highlight the main conclusions. Keeps the introduction comprehensible to scientists working outside the topic of the paper.
Response 1: We have shortened and structured the introduction according to your recommendations and separated it from the literature review.
Point 2: I suggest authors to put a new section after introduction that only explains the relationship between INDUSTRY 4.0 and research and development (R & D) projects.
Response 2: We added new paragraph in the literature review to explain relationship between R & D (as it has been defined by Fractati Manual of OBSE since 1963) and new phenomenon INDUSTRY 4.0, which has no classification in the international statistical systems and no aggregated data (neither on the national nor the international level).
Point 3: Briefly describe the limitations at the end of conclusion section.
Response 3: The limitations were added in the end of conclusion section.
Round 2
Reviewer 1 Report
I am in the pleasant situation to realize that my comments on the paper titled “Implementation of R & D results and industry 4.0 influenced by selected macroeconomic indicators” proved to be useful to the authors. I believe that the revised version of the paper is publishable to “Applied Sciences”.
Reviewer 2 Report
The authors have addressed the suggested remarks. In sum: accepted.